# The Combined Effects of Magnesium Oxide and Inulin on Intestinal Microbiota and Cecal Short-Chain Fatty Acids

**DOI:** 10.3390/nu13010152

**Published:** 2021-01-05

**Authors:** Kanako Omori, Hiroki Miyakawa, Aya Watanabe, Yuki Nakayama, Yijin Lyu, Natsumi Ichikawa, Hiroyuki Sasaki, Shigenobu Shibata

**Affiliations:** Laboratory of Physiology and Pharmacology, School of Advanced Science and Engineering, Waseda University, 2-2 Wakamatsu-cho, Shinjiku-ku, Tokyo 162-8480, Japan; k.omori@ruri.waseda.jp (K.O.); hgbbst-hiroki@toki.waseda.jp (H.M.); aya_watanabe7115@suou.waseda.jp (A.W.); yukibecky-6991@akane.waseda.jp (Y.N.); ikin@fuji.waseda.jp (Y.L.); natsu3@ruri.waseda.jp (N.I.); hiroyuki-sasaki@asagi.waseda.jp (H.S.)

**Keywords:** MgO, water-soluble dietary fiber, SCFA, intestinal flora, laxative, administration timing

## Abstract

Constipation is a common condition that occurs in many people worldwide. While magnesium oxide (MgO) is often used as the first-line drug for chronic constipation in Japan, dietary fiber intake is also recommended. Dietary fiber is fermented by microbiota to produce short-chain fatty acids (SCFAs). SCFAs are involved in regulating systemic physiological functions and circadian rhythm. We examined the effect of combining MgO and the water-soluble dietary fiber, inulin, on cecal SCFA concentration and microbiota in mice. We also examined the MgO administration timing effect on cecal SCFAs. The cecal SCFA concentrations were measured by gas chromatography, and the microbiota was determined using next-generation sequencing. Inulin intake decreased cecal pH and increased cecal SCFA concentrations while combining MgO increased the cecal pH lowered by inulin and decreased the cecal SCFA concentrations elevated by inulin. When inulin and MgO were combined, significant changes in the microbiota composition were observed compared with inulin alone. The MgO effect on the cecal acetic acid concentration was less when administered at ZT12 than at ZT0. In conclusion, this study suggests that MgO affects cecal SCFA and microbiota during inulin feeding, and the effect on acetic acid concentration is time-dependent.

## 1. Introduction

Constipation is a common condition that occurs in many people worldwide. According to an online survey conducted in Japan in 2014, 28.4% of men and women aged 20 to 79 years identified themselves as constipated [1]. A systematic survey of North America reported that the estimated constipation prevalence ranged from 12% to 19% and increased particularly among those aged 65 years and older [2]. Constipation not only reduces the patient’s quality of life but also leads to a poor life prognosis. In addition, constipation is a major financial burden [3].

Constipation is a cross-disciplinary disease that doctors in all departments recognize, but treatment is not straightforward. In Japan, the first chronic constipation practice guideline was published in 2017 [4]. The basic drug treatments suggested by the Guidelines were to use non-stimulant laxatives. A stimulant laxative was required when the effects of lifestyle-related guidance and non-stimulant laxatives were insufficient. Among non-stimulant laxatives, low-priced magnesium oxide (MgO) is often used as a first-line drug in Japan. However, since hypermagnesemia has been reported in patients with renal dysfunction [5], the Guidelines revised in 2020 require electrolyte monitoring in patients with renal failure, heart disease, electrolyte abnormalities, or those taking diuretics [6,7]. MgO is not only a laxative but also an antacid.

Magnesium oxide is classified as a saline laxative in the osmotic laxative group. When in the stomach, MgO reacts with gastric acid (HCl) to form magnesium chloride (MgCl_2_), which reacts with intestinally secreted pancreatic fluid (NaHCO_3_) to form poorly absorbable bicarbonates (magnesium bicarbonate: Mg(HCO_3_)_2_) and carbonates (magnesium carbonate: MgCO_3_). When the intestinal salt concentration increases, the osmotic pressure pulls water from the intestinal wall to the intestinal lumen in an attempt to maintain equal osmotic pressure on both sides of the membrane. This increases the amount of water inside the intestine and softens the stool.

The oral dosage of MgO as a laxative in Japan is 2 g per day for adults. It is administered three times a day, before or after meals, or once before bedtime. Although MgO pharmacokinetic studies are limited, a study using rats showed that approximately 15% of orally administered MgO is absorbed, and plasma magnesium concentration peaks about three hours after administration. Approximately 85% of orally administered MgO is excreted in the feces without being absorbed [8].

While there is a treatment for constipation with drugs, the Japanese guidelines for chronic constipation in 2020 states that lifestyle-related improvement is the first step in treatment, and the Ministry of Health, Labour, and Welfare recommends a daily dietary fiber intake of approximately 20 g/day for adult males and approximately 17 g/day for adult females [9]. The World Gastroenterology Organization guidelines also include a gradual increase in dietary fiber as one of the first steps in treating constipation [10].

There are as many as 100 trillion bacteria including more than 1000 bacterial species in the human intestine, which make up the gut microbiota [11]. The study by Qin J. et al. establishing a catalog of non-redundant human intestinal microbial genes from 124 European individuals shows that each individual harbors at least 160 prevalent bacterial species and about 536,000 prevalent genes. The study also indicates that about 38% of each individual’s genes were shared with at least half of the individuals of the study cohort [12]. Internal bacteria ferment dietary fiber to produce short-chain fatty acids (SCFAs), such as acetic acid, propionic acid, and butyric acid [13]. Acetic acid, propionic acid, and butyric acid account for 90–95% of SCFAs in the colon [14]. In the pathways for biosynthesis of SCFAs, lactic acid is converted to propionic acid, and also to pyruvate that is further converted to butyric acid, acetic acid and propionic acid [13,15,16].

There is a report that colonic pH and SCFA production after in vitro colonic fermentation are inversely related [17]. SCFAs produced by bacterial fermentation of dietary fiber create a more acidic environment in the colon. This environment is beneficial for the developing bacteria, and the proliferation of colonic bacteria increases stool bulk [18]. In other words, dietary fiber ingestion leads to an increase in intestinal SCFAs produced by intestinal bacteria, leading to constipation relief. The water-soluble dietary fiber, inulin, is known to increase SCFA levels in mice portal vein blood and cecal contents [19,20,21]. 

In addition, SCFAs not only regulate the intestinal environment and help relieve constipation, but they also induce regulatory T-cell differentiation [22], activate SCFA receptor, GPR43, to increase muscle and liver insulin sensitivity and to regulate energy balance [23], and may be associated with obesity [24]. Like these findings, SCFAs also play an important role in regulating other various biological functions.

In most organisms, there is a mechanism called a circadian clock that creates rhythmic cycles of biological function. The circadian clock locus in mammals is in the suprachiasmatic nucleus (SCN) of the hypothalamus and is called the central clock. Additionally, most organs and tissues have circadian clocks, called peripheral clocks [25]. Light is the most important stimulus for SCN. The peripheral clock is not only synchronized by the SCN, but also regulated by diet, medication, stress, and exercise [26,27,28,29,30]. Circadian clock disruption may lead to liver diseases, such as fatty liver and cirrhosis, as well as mood disorders, obesity, diabetes, and cancer [31,32,33]. Intestinal microbial rhythmicity is regulated by the circadian clock [34]. SCFAs produced by the gut flora modulate the peripheral clock phase in mice [35]. 

The study on effective medication timing in relation to the biological circadian rhythm is called chronopharmacology. Likewise, the study on effective nutrient intake timing in relation to the biological circadian rhythm is called chrononutrition. Previous studies have shown the importance of chronopharmacology and chrononutrition [36,37,38].

As noted, although many patients have used dietary fiber in combination with MgO to relieve constipation, no studies have examined the MgO effect on SCFA production from dietary fiber or its effect on the microbiota. And, no studies have investigated the effective MgO administration timing while ingesting dietary fiber.

This study aimed to investigate the MgO effects on SCFA and lactic acid concentrations in the cecum content and on the fecal microbiota diversity when combined with the water-soluble dietary fiber, inulin. We also examined the appropriate MgO administration timing.

## 2. Materials and Methods

### 2.1. Animals

Eight-week or nine-week-old male Institute of Cancer Research (ICR) mice (Tokyo Laboratory Animals Science Co., Ltd., Tokyo, Japan) were used. For all experiments, the mice were divided into groups of five or six mice and housed with population sizes of three or less. The mice were maintained at a room temperature 22 ± 2 °C, with 60 ± 5% relative humidity, using 100 to 150 lux light intensity, and on a 12-h light/12-h dark cycle with lights-on at 08:00 and lights-off at 20:00. Lights-on time was defined as Zeitgeber time 0 (ZT0) and lights-off time as 12 (ZT12). 

The procedures were conformed to the Japanese government laws and were approved by the Committee for Animal Experimentation of the School of Science and Engineering at Waseda University (Permission: 2018-A017, 2019-A058).

### 2.2. Experimental Procedure

#### 2.2.1. Experimental Design

Groups of mice and the experimental schedule of each experiment are shown in Figure 1.

In experiment one, we examined whether the combined use of inulin and MgO during normal feeding influenced cecal pH, lactic acid concentration, and SCFA concentrations. We prepared four kinds of diets: AIN-93M, AIN-93M containing 2.5% inulin, AIN-93M containing 2.5% inulin plus 0.125% MgO, or AIN-93M containing 2.5% inulin plus 0.25% MgO. AIN-93M is a maintenance purified diet for mice. In order to determine the appropriate MgO concentration for the following experiments, two different concentrations of MgO (0.125% and 0.25%) were used for the diet. Mice were under free-feeding and free-drinking water conditions. Mice were housed under the noted conditions for 11 days and sampled at ZT4. The cecal pH was measured and the cecal contents were collected.

In experiment two, we examined whether the combined use of inulin and MgO during a high-fat diet (HFD) feeding influenced cecal pH, lactic acid concentration, SCFA concentrations and microbiota. Magnesium oxide is a remedy for constipation. It is also known that constipation deteriorates the intestinal environment. Since HFD disturbs the intestinal environment [39], HFD was used in experiment two. A previous study in our laboratory had confirmed that HFD deteriorates the intestinal environment [40]. Then, the cecal pH, lactic acid concentration, and SCFA concentration in the cecum were measured, and the microbiota was analyzed. In the mixed diet, the concentration for inulin and MgO were 2.5% and 0.25%, respectively. To adjust the total dietary fiber amount in the chow, the control group was fed 2.5% of cellulose, which is insoluble dietary fiber. All groups were under free-feeding and free-drinking water conditions. The mice were housed under the noted conditions for 11 days and then sampled at ZT4 to determine the cecal pH and to collect the cecal contents and feces.

In experiment three, we investigated the effects of MgO administration timing on the lactic acid and SCFA concentrations. Mice fed 2.5% inulin in the HFD was used as the inulin group, and mice fed 2.5% cellulose in the HFD was used as the control group. In the inulin plus MgO group, a HFD containing 2.5% inulin was fed, and MgO (250 mg/kg BW) was orally administered at ZT0 or ZT12 for 11 days. Cecal contents and feces samples were collected on the last day four hours after MgO was administered. All mice were under free-feeding and free-drinking water conditions. Mice were housed under the noted conditions for 11 days and sampled at ZT4 or ZT16.

#### 2.2.2. Measuring the Cecal pH

The cecal pH was measured by inserting the glass tipped pH meter electrode (Eutech Instruments, Vernon Hills, IL, USA) directly into the cecum immediately after collection.

#### 2.2.3. Measuring the SCFA and Lactic Acid Levels in the Cecum

Cecal SCFA and lactic acid concentrations were measured using gas chromatography with a flame ionization detector (Shimadzu Co., Kyoto, Japan) as described in a previous report [41]. First, 50 mg of the cecal contents were measured in a 1.5 mL tube, and 600 μL of a diethyl ether and ethanol mixture at a ratio of 2:1 was added. Then, 50 μL of sulfuric acid was added and vortexed. The mixture was centrifuged at 14,000× *g* for 30 s at room temperature. The supernatant was injected into the capillary column (InertCap Pure-WAX 30 m × 0.25 mm, df = 0.5 μm; GL Sciences, Tokyo, Japan). The initial temperature was 80 °C and the final temperature was 200 °C. Helium was used as the carrier gas. After supernatant removal, the cecal contents were thoroughly dried and weighed. Cecal SCFA concentrations were calculated as μmol/mg of dry cecal weight. This procedure of quantitative SCFA and lactic acid analysis was established in our papers [37,38,40].

#### 2.2.4. Fecal DNA Extraction 

Fecal DNA was extracted according to a previous report [42]. First, 200 mg of mouse feces and 20 mL of phosphate-buffered saline (PBS) were added to 50 mL Falcon tubes and sufficiently suspended with a spatula and vortexing. The suspension was filtered through a 100 μm nylon mesh filter (Corning Inc., New York, NY, USA). Then, 10 mL of fresh PBS was added to each 50 mL Falcon tube. The tubes were washed thoroughly, and the solutions were also filtered. The obtained filtrates were centrifuged at 9000× *g* for 20 min at 4 °C and the supernatant was removed. The pellets were resuspended in 1.5 mL of 10 mM Tris-HCl (FUJIFILM Wako Pure Chemical Co., Osaka, Japan) and 10 mM EDTA buffer, and centrifuged at 10,000× *g* for 5 min at 4 °C. The supernatant was removed and the pellet was resuspended in 800 μL of 10 mM Tris-HCl and 10 mM EDTA buffer. Next, 100 μL of lysozyme (FUJIFILM Wako Pure Chemical Co., Osaka, Japan) solution (150 mg lysozyme in 1 mL of 10 mM Tris-HCl, and 10 mM EDTA) was added, mixed by inversion, and incubated at 37 °C for 1 h. Following, 20 μL of acromopeptidase (FUJIFILM Wako Pure Chemical Co., Osaka, Japan) solution (100 units/μL) was added, mixed by inversion, and incubated at 37 °C for 30 min. Then, 50 μL of the proteinase K solution (Promega Co., Madison, WI, USA) and 20% SDS solution were added, mixed by inversion, and incubated at 55 °C for 1 h. An equal volume of phenol/chloroform/isoamyl alcohol (PCI) (Invitrogen, Carlsbad, CA, USA) solution was added and mixed sufficiently until the solution became white. The suspension was centrifuged at 6000× *g* for 10 min at room temperature. The supernatant was transferred to a new 2 mL tube. The PCI solution and the subsequent steps were repeated twice. Then, 100 μL of a 3 M sodium acetate solution and 900 μL of isopropanol (FUJIFILM Wako Pure Chemical Co., Osaka, Japan) were added to the supernatant. The suspension was centrifuged at 6000× *g* for 10 min at 20 °C and the supernatant was removed. Then, 1 mL of cold 70% ethanol was added to the DNA pellets, and the suspension was centrifuged at 15,000× *g* for 5 min to remove the supernatant. Then, 500 μL of cold 70% ethanol was added, and the suspension was centrifuged at 15,000× *g* for 5 min to remove the supernatant. The pellets were air-dried until it became translucent. To the pellet, 99 μL of TE buffer (10 mM Tris-HCl, and 1 mM EDTA) and 1 μL of RNase (10 μg/mL) (FUJIFILM Wako Pure Chemical Co., Osaka, Japan) were added. The suspension was kept warm at 37 °C overnight. Equal amounts of 20% PEG solution (Tokyo Chemical Industry Co., Ltd., Tokyo, Japan) were added to the suspension and were allowed to stand on ice for 10 min. The supernatant was removed by centrifugation at 10,000× *g* for 10 min at 4 °C. The DNA was rinsed using 500 μL of cold 70% ethanol, centrifuged at 15,000× *g* for 5 min, and the supernatant was removed. The pellet was peeled off the tube by tapping, 500 μL of cold 70% ethanol was added, and the suspension was centrifuged at 15,000× *g* for 5 min to remove the supernatant. The final DNA was air-dried, resuspended in 40 μL of TE buffer, and stored at −20 °C.

#### 2.2.5. 16S rDNA Sequencing

Gut flora 16S rDNA was treated according to the Illumina Miseq System protocol, *16S Metagenomic Sequencing Library Preparation*. The V3-V4 variable region of the 16S rDNA gene was amplified by the polymerase chain reaction (PCR) using the primers described below:

Forward Primer = 5′-TCGTCGGCAGCGTCAGATGTGTATAAGAGACAGCCTACGGGNGGCWGCAG-3′

Reverse Primer = 5′-GTCTCGTGGGCTCGGAGATGTGTATAAGAGACAGGACTACHVGGGTATCTAATCC-3′

The PCR amplification was performed with 2.5 µL of microbial DNA (5 ng/µL), 5 µL of Forward Primer (1 µmol/L), 5 µL of Reverse Primer (1 µmol/L), and 12.5 µL of 2× KAPA HiFi HotStart Ready Mix (Kapa Biosystems Inc., Wilmington, MA, USA). The PCR reaction conditions were as follows: After being kept at 95 °C for 3 min, a cycle of 95 °C for 30 s, 55 °C for 30 s, and 72 °C for 30 s was repeated for 25 cycles. A final extension was performed at 72 °C for 5 min. The PCR products were purified using AM PureXP beads (Beckman Coulter Inc., Sacramento, CA, USA). Index PCR was performed by adding 5 µL of purified amplicon PCR product to 5 µL of Nextera XT Index Primer 1 (Illumina Inc., Hayward, CA, USA), 5 µL of Nextera XT Index Primer 2 (Illumina Inc., Hayward, CA, USA), 25 µL of 2× KAPA HiFi HotStart ReadyMix (Kapa Biosystems, Wilmington, MA, USA), and 10 µL of PCR-grade water. The reaction conditions for the index PCR were as follows: After keeping at 95 °C for 3 min, a cycle of 95 °C for 30 s, 55 °C for 30 s, and 72 °C for 30 s was repeated for 8 cycles. A final extension was performed at 72 °C for 5 min. Index PCR products were purified using AMPure XP beads (Beckman Coulter Inc., Brea, CA, USA). The purified DNA quality was confirmed using an Agilent 2100 Bioanalyzer (Agilent Technologies Inc., Santa Clara, CA, USA) and DNA1000 kit. Finally, the DNA library concentration was adjusted to 4 nmol/L. The DNA library was sequenced on the Illumina MiSeq 2 × 300 bp platform using the MiSeq Reagent Kit v3 (Illumina Inc., San Diego, CA, USA). This sequence was performed according to the manufacturer’s instructions.

#### 2.2.6. Analysis of 16S rDNA Gene Sequence

The 16S rDNA extracted from feces was analyzed by the quantitative insights to microbiological ecology (QIIME) pipeline, version 1.9.1 [43]. Reads after quality checks were classified by UCLUST as operational taxonomic units of 97% similarity [44]. These reads were compared with the Greengenes database Reference Sequence (August 2013 version). A total of 383,440 reads were obtained from 16 samples, with an average of 23,965 ± 591.2728 reads per sample. QIIME analyzed these reads and calculated β-diversity and the taxonomy from phylum to species levels. β-Diversity was shown in the principal coordinate analysis (PCoA). The PCoA was calculated using unweighted UniFrac distances. A Simpson Index was calculated from the taxonomic composition summary and the α-diversity results were presented.

#### 2.2.7. Statistical Analysis

All data except for β-diversity are shown as mean ± standard error and analyzed by GraphPad Prism (version 8.4.3, GraphPad Software Inc., San Diego, CA, USA). First, the presence or absence of normality was determined by the Kolmogorov-Smirnov test. Next, the equivariance was determined by Bartlett’s test. When these analyses showed normality and equivariance, a one-way analysis of variance (ANOVA) tests using Tukey post hoc analysis was performed. Kruskal-Wallis test using Dunn’s post hoc test was performed if there was no normality or unequal variance. The *p*-values shown were adjusted for multiple comparison. β-Diversity was assessed by permutational multivariate analysis of variance (PERMANOVA), and PERMANOVA was analyzed using QIIME.

## 3. Results

### 3.1. Effects of Inulin and MgO Combination on Normal Intestinal Environments

#### 3.1.1. The Combined Inulin and MgO Effects on Cecal pH

There were no significant differences in cecal pH among the four groups, but changes in pH were observed when mice were subjected to inulin and MgO intake. Specifically, inulin feeding lowered the cecal pH. However, the combined inulin and MgO treatment resulted in increased cecal pH compared with the inulin group. Cecal pH comparison of the inulin plus 0.125% MgO group and the inulin plus 0.25% MgO group showed that the cecal pH was slightly higher in the 0.25% MgO-added group (Figure 2a). These results suggest that MgO may increase cecal pH in a dose-dependent manner.

#### 3.1.2. The Combined Inulin and MgO Effects on Cecal SCFAs 

Inulin intake significantly increased lactic acid concentration in the cecal contents. Interestingly, the inulin plus 0.125% MgO group had decreased lactic acid concentration, and lactic acid concentration further significantly decreased in the inulin plus 0.25% MgO group compared with the inulin group. The lactic acid concentration was lower in the inulin plus 0.25% MgO group than in the 0.125% MgO-added group, but no significant difference was observed (Figure 2b).

Although no significant difference was observed, inulin increased acetic acid concentration. Compared with the control group, the acetic acid concentration was significantly higher in the inulin plus 0.125% MgO group. On the other hand, the inulin plus 0.25% MgO group has lower acetic acid concentration than the 0.125% MgO-added group (Figure 2c). This suggests that MgO may reduce acetic acid concentration in a dose-dependent manner when used in combination with inulin.

The propionic acid concentration was significantly higher in the inulin group, the inulin plus 0.125% MgO group, and the inulin plus 0.25% MgO group when compared with the control group (Figure 2d).

A significantly higher concentration of butyric acid was detected in the inulin group and the inulin plus 0.125% MgO group compared with the control group. No significant difference was observed between the control and inulin plus 0.25% MgO groups (*p* = 0.1705). Although no significant difference was observed, the 0.25% MgO-added group showed lower butyric acid concentration compared with the 0.125% MgO-added group (Figure 2e).

The lactic acid and total SCFA concentration sum was significantly higher in the inulin group and the inulin plus 0.125% MgO group compared with the control group. Although not significantly different, the lactic acid and total SCFA concentration sum was lower in the inulin plus 0.125% MgO group compared with the inulin group, and was lower in the inulin plus 0.25% MgO group compared with the 0.125% MgO-added group (Figure 2f).

### 3.2. The Combined Inulin and MgO Effects on the Intestinal Environment and Microbiota during HFD Feeding 

Magnesium oxide may be used in conditions where the intestinal environment has deteriorated. In experiment two, the combined inulin and MgO effects on the intestinal environment were investigated using a HFD, which is known to disturb the intestinal environment in mice [38,40]. Since experiment one suggested dose-dependent MgO effects in a normal diet, MgO was mixed in the diet at a concentration of 0.25% for experiment two (Figure 1b).

#### 3.2.1. The Combined Inulin and MgO Effects on Cecal pH during HFD Feeding

The cecal pH was significantly reduced in the inulin group compared with the control group. However, the cecal pH increased significantly in the inulin plus MgO group compared with the inulin group (Figure 3a). These results suggest that MgO increases the cecal pH that was decreased by inulin.

#### 3.2.2. The Combined Inulin and MgO Effects on Cecum SCFAs during HFD Feeding

The lactic acid concentration was significantly increased by inulin. Although the difference was not significant (*p* = 0.3697), the combined use with MgO reduced the lactic acid concentration (Figure 3b). Compared with the control group, the inulin group showed a higher acetic acid concentration. However, the inulin plus MgO group showed a significantly lower acetic acid concentration compared with the control group and inulin group (Figure 3c). The propionic acid concentration increased significantly with inulin intake, whereas it significantly decreased with the combined with MgO compared with the inulin group (Figure 3d). The butyric acid concentration also increased with inulin intake and significantly decreased with combined use of MgO compared with the inulin group (Figure 3e). The lactic acid and total SCFA concentration sum was significantly increased in the inulin group. The sum was significantly decreased with combined use of MgO compared with the inulin and control groups (Figure 3f). These results suggest that inulin increases lactic acid, acetic acid, propionic acid, and butyric acid concentrations, while the combined use of MgO reduces the inulin-induced increased concentrations.

#### 3.2.3. The Combined Inulin and MgO Effects on the Microbiota during HFD Feeding

Microbiota β-diversity analysis showed significant differences in microbiota composition among the three groups (Figure 4a). Permutational multivariate analysis of variance was performed using two of the three groups to determine the significantly different group. First, a comparison between the control group and the inulin group showed that inulin feeding significantly altered the composition of the microbiota (Figure 4b). In addition, a significant microbiota composition change was observed between the inulin and the inulin plus MgO group (Figure 4c). There was also a significant microbiota composition change between the control group and the inulin plus MgO group (Figure 4d). These results indicated that inulin alone and inulin plus MgO significantly altered the microbiota composition. Also, the microbiota change associated with the combination of inulin and MgO differed from the control group microbiota. Furthermore, we presented the microbiota taxonomic summary and expressed α-diversity by Simpson index and Chao1 (Figure 5). There was no significant difference in Simpson index or Chao1 between the inulin and the inulin plus MgO groups. (Figure 5b,c).

We examined whether there was a change in each bacterium at the genus level (Figure 6). The relative abundance of *Coprococcus*, *Lactococcus*, *Parabacteroides*, and *Streptococcus* was significantly reduced in the inulin group compared with the control group. The relative abundance of these bacteria reduced by inulin was increased with the combined MgO use, but the elevation was not significantly different. Additionally, the relative abundance of *Ruminococcus*, *Odoribacter*, *Oscillospira*, and *Ruminococcus* was significantly reduced in the inulin plus MgO group compared with the control group. The relative abundance of *Turicibacter* was significantly elevated in the inulin group compared with the control group. However, the MgO combination group severely decreased the *Turicibacter* relative abundance level. The relative abundance of *Butyricimonas* significantly decreased in the inulin group and MgO combination group compared with the control group. The relative abundance of *Lactobacillus* was elevated in inulin group. The *Lactobacillus* relative abundance in the inulin-MgO combination group was decreased compared with the inulin group.

### 3.3. The Relationship between the MgO Effect on the Intestinal Environment and Administration Timing 

Experiments one and two have shown that MgO alters the intestinal environment and microbiota during inulin feeding. In experiment three, we investigated whether there was a relationship between the MgO administration time and its effects. Magnesium oxide was orally administered at ZT0, the beginning of the inactive phase, or at ZT12, the beginning of the active phase, and sampling was carried out four hours after each oral administration (Figure 1c). 

In the group sampled at ZT4, the cecal pH in the inulin group decreased but not significantly. In the group sampled at ZT16, inulin significantly reduced the cecal pH. Magnesium oxide significantly increased cecal pH at both sampling times (Figure 7a). Regardless of the MgO administration time, the lactic acid concentration increased in the inulin group compared with the control group and decreased when combined with MgO (Figure 7b,c). The acetic acid concentration was significantly reduced by MgO in the ZT4 sampling group. On the other hand, in the ZT16 sampling group, no significant difference was observed among all groups (Figure 7d,e). The propionic acid concentration was significantly increased by inulin in the ZT4 and ZT16 sampling groups. Although not a significant difference, MgO showed a decrease in the propionic acid concentration increased by inulin at both sampling times (Figure 7f,g). In both the ZT4 and ZT16 sampling groups, MgO significantly reduced the butyric acid concentration that was increased by inulin (Figure 7h,i). Magnesium oxide significantly decreased the sum of the lactic acid and SCFA concentrations that were increased by inulin in the ZT4 sampling group. In the ZT16 sampling group, MgO tended to decrease the sum concentration increased by inulin, but no statistically significant difference was observed (Figure 7j,k). These results suggest that MgO administration at ZT0 had a stronger effect on the acetic acid concentration than MgO administration at ZT12.

## 4. Discussion

### 4.1. Combined Inulin and MgO Effect on the SCFA Concentrations in the Cecum

The combined inulin and MgO effect on the SCFA concentrations in the cecum was examined. It was shown that MgO reduced inulin-induced increased lactic acid and SCFA concentrations regardless of the diet-type or MgO administration method (Figure 2, Figure 3 and Figure 7). Short-chain fatty acids not only regulate the intestinal environment but are involved in systemic physiological functions [22,23] and also act as anti-inflammatory substances. In the alcoholic liver disease model mice fed with inulin, SCFAs improved inflammation [45]. Giving drinking water that contains acetic acid to mice before or during the inflammatory arthritis induction reduced paw swelling [46]. It has also been reported that allergic airway inflammation is suppressed in mice fed propionic acid [47]. Butyric acid regulates intestinal inflammation by activating histone deacetylase inhibition, enhancing Foxp3 gene promoter acetylation, and promoting Treg-cell differentiation [22]. Finally, lactic acid from *Lactobacillus johnsonii* plays a protective role in small bowel injury induced by indomethacin, a non-steroidal anti-inflammatory drug [48].

SCFAs contribute to maintaining systemic immune homeostasis, and therefore SCFA reduction may induce or exacerbate the above-mentioned diseases. In this study, MgO ingestion reduced cecal SCFA levels that were increased by inulin. This suggests that MgO may suppress the anti-inflammatory SCFA effects and induce or exacerbate inflammatory diseases. It would be interesting to confirm whether MgO exacerbates the inflammation using a pathological model of each inflammatory disease.

### 4.2. Changes in Cecal pH and Intestinal Flora Composition

In experiment two using mice fed with a HFD, the cecal pH decreased significantly due to inulin feeding. It was increased significantly when combined with MgO (Figure 3a). Furthermore, the intestinal flora composition was significantly changed in the inulin plus MgO group compared with the inulin group (Figure 4c).

Both rabeprazole and vonoprazan are known to act on proton pumps to suppress gastric acid secretion, but their mechanisms of action are different. Additionally, both of these agents significantly alter bacterial flora composition in mice small intestine and colon [49]. This result suggests that the intestinal flora composition changes as the stomach acidity affects the intestinal pH. In this study as well, cecal pH increased and intestinal flora composition changed. MgO is also used as an antacid, so these data may indicate that MgO administration increased the gastric pH and intestinal pH, and changed the intestinal flora composition accordingly. In experiment two, the inulin-induced increased SCFA and lactic acid concentrations were decreased when combined with MgO (Figure 3b–f). One of the causes of these phenomena may be the intestinal flora composition changes due to the increased intestinal pH.

Magnesium oxide is converted to bicarbonate in the intestine and increases intestinal osmotic pressure, attracting water to the intestinal cavity, and softening the intestinal contents. Polyethylene glycol, an osmotic laxative, alters the gut flora over the long term [50]. This may indicate that intestinal osmotic pressure changes may also be involved in intestinal flora changes. 

### 4.3. Microbiota α-Diversity

In this study, we compared the α-diversity of the control group, inulin group, and inulin plus MgO group, but no significant difference was found between the inulin and inulin plus MgO groups (Figure 5b,c). A previous study reported that α-diversity was significantly lowered in proton pump inhibitor (PPI) users [51]. Sun Min Lee reported that long-term PPI administration significantly reduced the intestinal flora α-diversity in 2-year-old rats, while no significant α-diversity reduction was seen in 74-week-old rats [52]. In other words, changes in α-diversity due to PPI administration occurred only in older rats. In our study, MgO did not change the intestinal flora α-diversity (Figure 5b,c). In our study, 8–9-week-old mice were used, and the results are consistent with the results of a previous study.

Magnesium oxide is used as a laxative not only for chronic constipation but also for constipation that occurs as a side effect due to cancer palliative treatment. Since many patients with cancer or constipation are elderly, further study in aged mice to examine the MgO effect on intestinal flora will be required. Examining the relationship between the MgO effect on the intestinal flora and mouse age will lead to age-appropriate clinical treatment regimes.

### 4.4. Combined Inulin and MgO Effect on Intestinal Bacteria at the Genus Level

A previous study has reported that the administration of a PPI increases the *Parabacteroides* and *Streptococcus* abundance ratio and decreases the *Ruminococcus* abundance ratio in human feces [51]. In another study, short-term PPI administration to patients with gastroesophageal reflux disease increased the *Streptococcus* abundance ratio and decreased the *Ruminococcus* abundance ratio in feces [53]. In experiment two, it was shown that the *Parabacteroides* and *Streptococcus* abundance ratio increased, and the *Ruminococcus* abundance ratio decreased in the MgO combination group compared with the inulin group (Figure 6). These results were similar to the bacterial abundance ratio fluctuation during PPI administration. Since MgO has an antiacid effect, the bacterial abundance ratio fluctuation may be similar to the fluctuations during PPI administration. 

On the other hand, the results were different from PPI administration regarding the *Lactobacillus* abundance ratio in feces. In experiment two, the *Lactobacillus* abundance ratio in the MgO combination group decreased compared with the inulin group (Figure 6). However, the *Lactobacillus* abundance ratio in feces increased with PPI administration in the previous studies [51,54]. In another paper, gastric acid secretion inhibitors, rabeprazole and vonoprazan, significantly increased the *Lactobacillus* abundance ratio in the colon, while neither 20 mg rabeprazole nor 20 mg vonoprazan increased *Lactobacillus* in the small intestine [49]. These results indicate that the effect of gastric pH changes on the intestinal flora depends on the regions along the gastrointestinal tract. Small intestine, which is close to the stomach, may be more affected by the decrease in gastric pH than the colon. 

In our study, fecal intestinal flora was examined and a decrease in the *Lactobacillus* abundance ratio was confirmed. Our results are consistent with the results of a previous study observed in the small intestine when rabeprazole or vonoprazan was administered [49]. These results suggest that MgO may strongly or over a long time increase the intestinal pH through its antacid action in the stomach, and affect the intestinal flora more strongly than PPI administration. In experiments one to three, the cecal lactic acid concentration increased by inulin decreased when inulin was combined with MgO. This may be because MgO reduced the *Lactobacillus* abundance ratio, a lactic acid-producing bacterium.

### 4.5. MgO Administration Timing Effects on Cecal pH and Cecal SCFAs

Circadian clock disruption may lead to liver diseases, such as fatty liver and cirrhosis, as well as mood disorders, obesity, diabetes, and cancer [30,31,32]. SCFA and L-lactic acid administration at ZT5 advances the circadian clock phase in mice kidney, liver, and submandibular glands [34]. In our study, inulin intake increased SCFA and lactic acid concentrations in the cecum, but the combined use with MgO decreased them. This result suggests that MgO may counteract the beneficial SCFA effects increased by inulin feeding. On the other hand, a previous study reported no phase change by SCFA and L-lactic acid administered at ZT0, ZT12, and ZT17 [34]. The SCFA effects on the circadian clock may be time-dependent. In the future, it will be necessary to pursue the appropriate timing of the combined dietary fiber and MgO administration, considering the SCFA effects on the circadian clock.

Since the onset of efficacy of some drugs is closely related to the circadian rhythm, it is very important to verify the appropriate administration timing for effective drug treatment [35].

In experiment three of this study, two different time points for MgO administration, ZT0, and ZT12, were used. We examined whether the combined inulin and MgO effect on cecal SCFA concentrations was time-dependent. Acetic acid in the cecum was significantly reduced when MgO was administered at ZT0, whereas no significant decrease was observed when MgO was administered at ZT12 (Figure 7d,e). This result indicated that the MgO effect on cecal acetic acid concentration was more prominent at ZT0 than at ZT12. This suggests that MgO administration at ZT12 affects cecal SCFA concentrations less than at ZT0. Since ZT0 is the start time of the inactive phase and ZT12 is the start time of the active phase, the administration at ZT12 is comparable to human administration when waking up. When used as a laxative by adults, the MgO dosage and administration procedure in Japan is to be orally given at 2 g/day in three times a day, before or after meals, or once before bedtime. Considering the effects on intestinal SCFAs, MgO administration when waking up for the day may be more appropriate. SCFAs are involved in systemic physiological functions, such as immune homeostasis maintenance, metabolism, and circadian clock synchronization [21,22,23,34,45,46,47]. Therefore, it is important to evaluate the administration timing considering the effect on SCFAs.

In the current experiments, we used HFD as a deterioration of the intestinal environment. However, in future experiments, we should confirm these results under impaired microbiota using constipation mouse model created by a drug such as loperamide.

## 5. Conclusions

In summary, this study showed that MgO increased cecal pH, as well as decreased cecal SCFA and lactic acid levels during inulin feeding. MgO significantly changed the intestinal flora composition. Furthermore, the MgO effect on the cecal acetic acid concentration was less when administered at ZT12 compared with ZT0. This is the first study to show that the combination of inulin and MgO affects cecal SCFAs, cecal lactic acid and intestinal flora. In addition, we are the first to show the MgO appropriate administration timing related to the effect on SCFAs in the cecum. It has been reported that intestinal flora disorders are associated with various diseases. Therefore, drug administration effects on the intestinal flora should be considered for systemic health management. In addition, since SCFAs are involved in systemic physiological functions and circadian rhythm, the drug administration timing should be considered for its effects on SCFAs.

## Figures and Tables

**Figure 1 nutrients-13-00152-f001:**
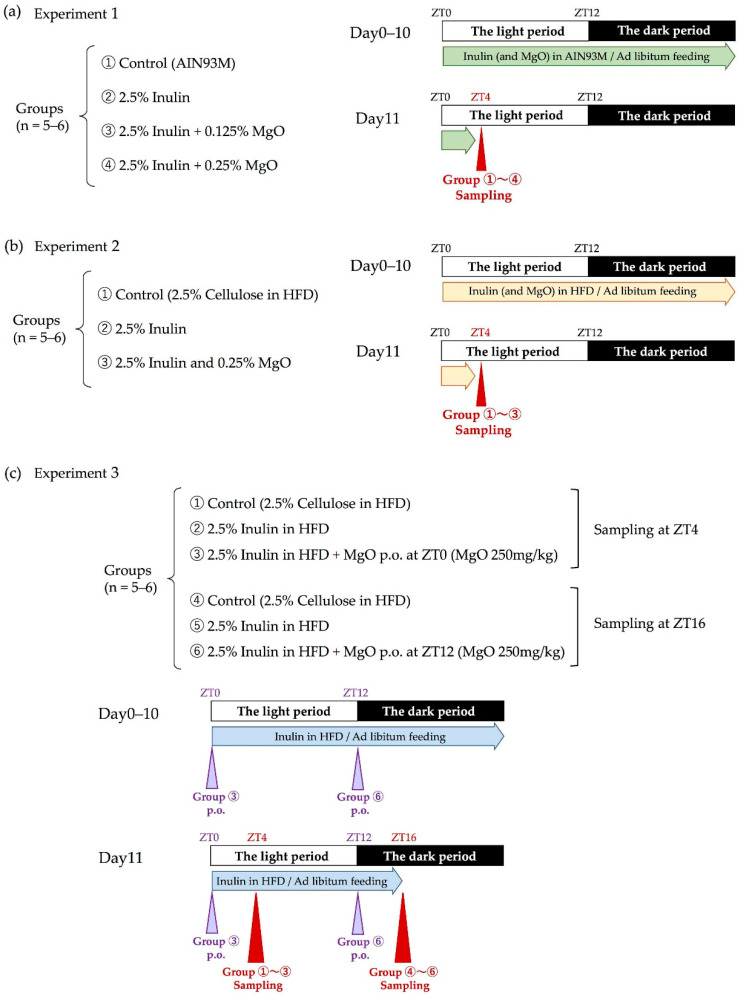
Experimental design. (**a**) Groups and schedule of experiment 1. (**b**) Groups and schedule of experiment 2. (**c**) Groups and schedule of experiment 3. White and black bars express a 12-h light/12-h dark cycle. ZT0 is lights-on time and ZT12 is lights-off time. Triangle arrows in red indicate sampling time. Triangle arrows in purple indicate oral administration time. Green, yellow or blue arrows indicate a period of ad libitum feeding. ZT: Zeitgeber time. MgO: magnesium oxide. HFD: high-fat diet.

**Figure 2 nutrients-13-00152-f002:**
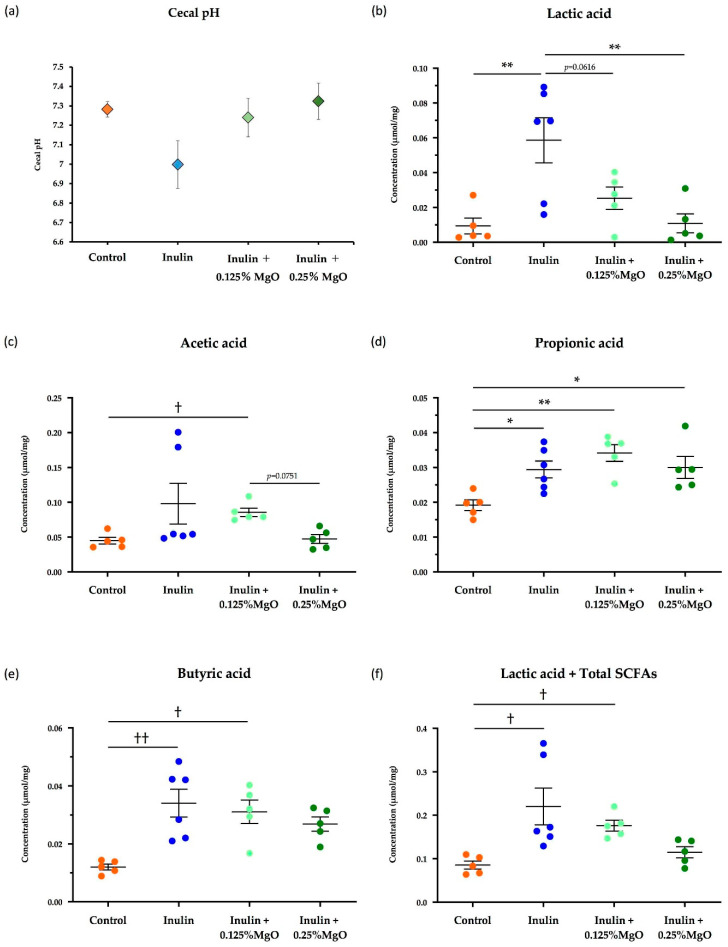
The combined inulin and MgO effects on cecal pH and SCFAs in mice on a normal diet. (**a**) Cecal pH. (**b**) Lactic acid concentration of the cecal content. (**c**) Acetic acid concentration of the cecal content. (**d**) Propionic acid concentration of the cecal content. (**e**) Butyric acid concentration of the cecal content. (**f**) Lactic acid concentration + total SCFA concentration of the cecal contents (Total SCFA concentration is the total sum of the acetic acid concentration, propionic acid concentration, and butyric acid concentration). AIN93-M was used as the diet, and inulin was mixed with the diet at a ratio of 2.5% in the inulin group and the MgO combination group. Magnesium oxide was given in the diet at a ratio of 0.125% or 0.25%. Concentrations of lactic acid and each SCFA were calculated per dry weight of the cecal contents. All data are expressed as mean ± standard error (Each group *n* = 5–6). * *p* < 0.05 and ** *p* < 0.01, evaluated using the one-way ANOVA with Tukey’s post hoc test. ^†^
*p* < 0.05, ^††^
*p* < 0.01, evaluated using the Kruskal-Wallis test with Dunn’s post hoc test. MgO: magnesium oxide. SCFA: short-chain fatty acid.

**Figure 3 nutrients-13-00152-f003:**
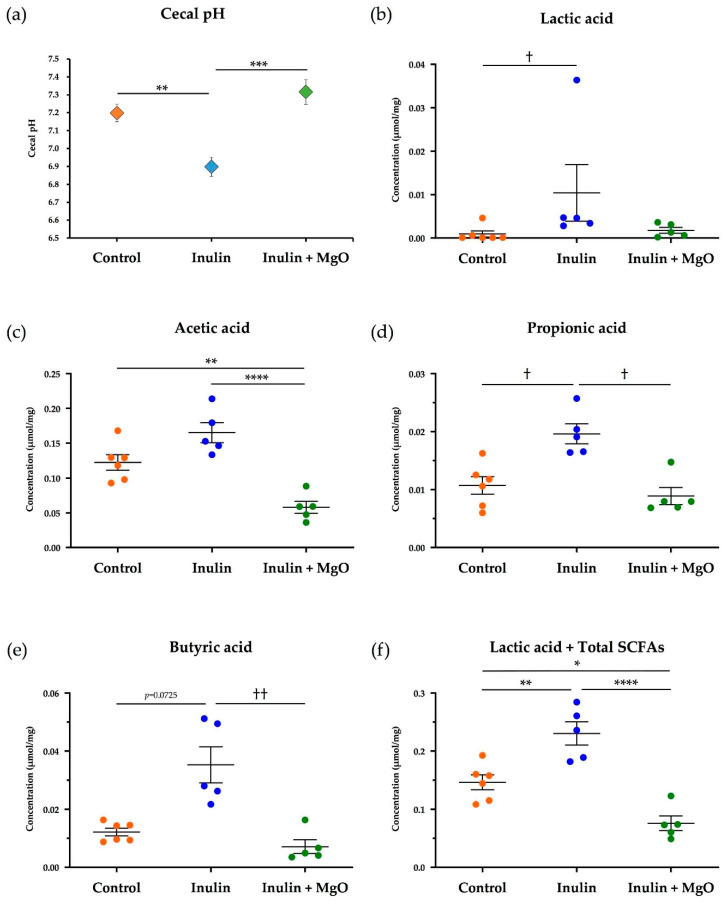
The combined inulin and MgO effects on cecal pH and SCFAs in mice on a HFD. (**a**) Cecal pH. (**b**) Lactic acid concentration of the cecal content. (**c**) Acetic acid concentration of the cecal content. (**d**) Propionic acid concentration of the cecal content. (**e**) Butyric acid concentration of the cecal content. (**f**) Lactic acid concentration + total SCFA concentration of the cecal contents (Total SCFA concentration is the total sum of the acetic acid concentration, propionic acid concentration, and butyric acid concentration). Inulin was mixed with the HFD at a ratio of 2.5% in the inulin group and the MgO combination group. Magnesium oxide was given in the diet at a ratio of 0.25%. Concentrations of lactic acid and each SCFA were calculated per dry weight of the cecal contents. All data are expressed as mean ± standard error (Each group *n* = 5–6). * *p* < 0.05, ** *p* < 0.01, *** *p* < 0.001, **** *p* < 0.0001, evaluated using the one-way ANOVA with Tukey’s post hoc test. ^†^
*p* < 0.05, ^††^
*p* < 0.01, evaluated using the Kruskal-Wallis test with Dunn’s post hoc test. MgO: magnesium oxide. SCFA: short-chain fatty acid. HFD: high-fat diet.

**Figure 4 nutrients-13-00152-f004:**
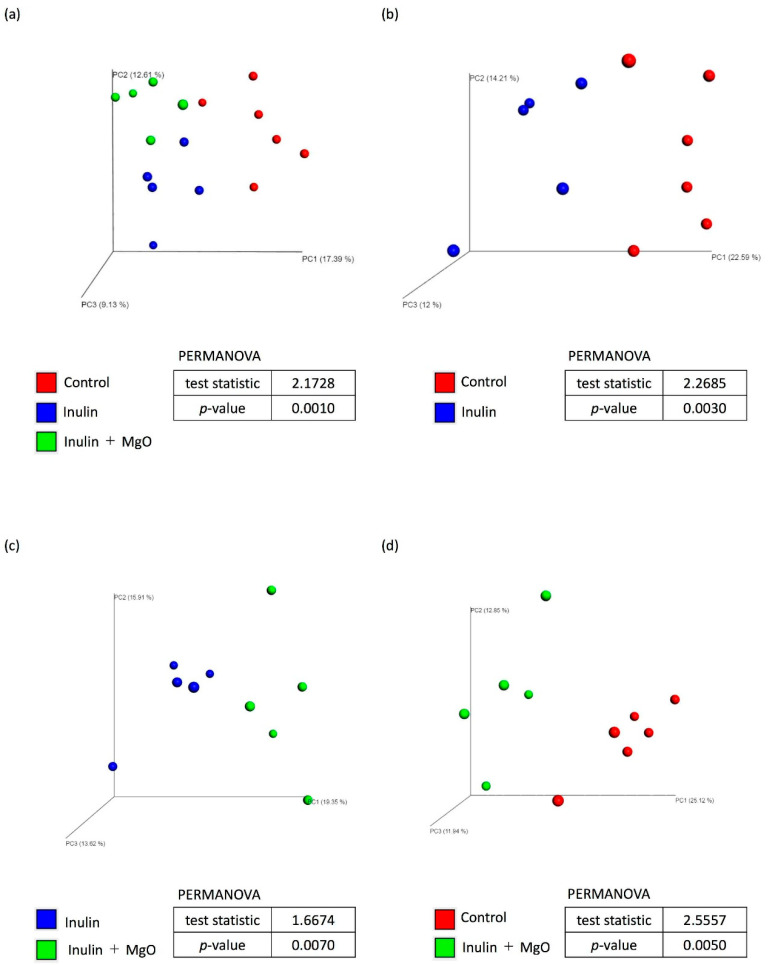
The combined inulin and MgO effects on microbiota β-diversity in mice on a HFD. (**a**) Control group vs. inulin group vs. inulin + MgO group. (**b**) Control group vs. inulin group. (**c**) Inulin group vs. inulin + MgO group. (**d**) Control group vs. inulin + MgO group. Inulin was mixed with the HFD at a ratio of 2.5% in the inulin group and the MgO combination group. Magnesium oxide was given in the diet at a ratio of 0.25%. PCoA plots were calculated using unweighted UniFrac distance. (Each group *n* = 5–6). All data in tables were analyzed by a permutational multivariate analysis of variance (PERMANOVA). MgO: magnesium oxide. HFD: high-fat diet.

**Figure 5 nutrients-13-00152-f005:**
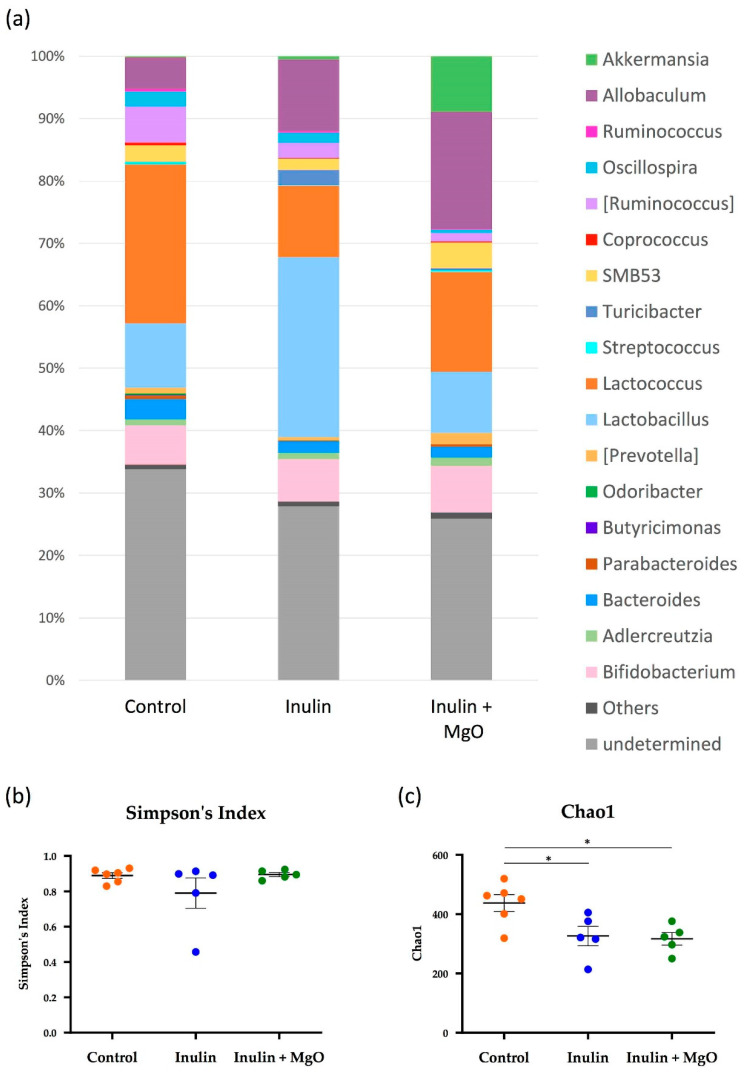
The combined inulin and MgO effects on microbiota α-diversity in mice on a HFD. (**a**) The taxonomic summary at the genus level. (**b**) Alpha-diversity measured by Simpson’s Index. (**c**) Alpha-diversity measured by Chao1. Inulin was mixed with the HFD at a ratio of 2.5% in the inulin group and the MgO combination group. Magnesium oxide was given in the diet at a ratio of 0.25%. All data are expressed as mean ± standard error (Each group *n* = 5–6). Data of Simpson’s Index were evaluated using the Kruskal-Wallis test with Dunn’s post hoc test. * *p* < 0.05, evaluated using the one-way ANOVA with Tukey’s post hoc test. MgO: magnesium oxide. HFD: high-fat diet.

**Figure 6 nutrients-13-00152-f006:**
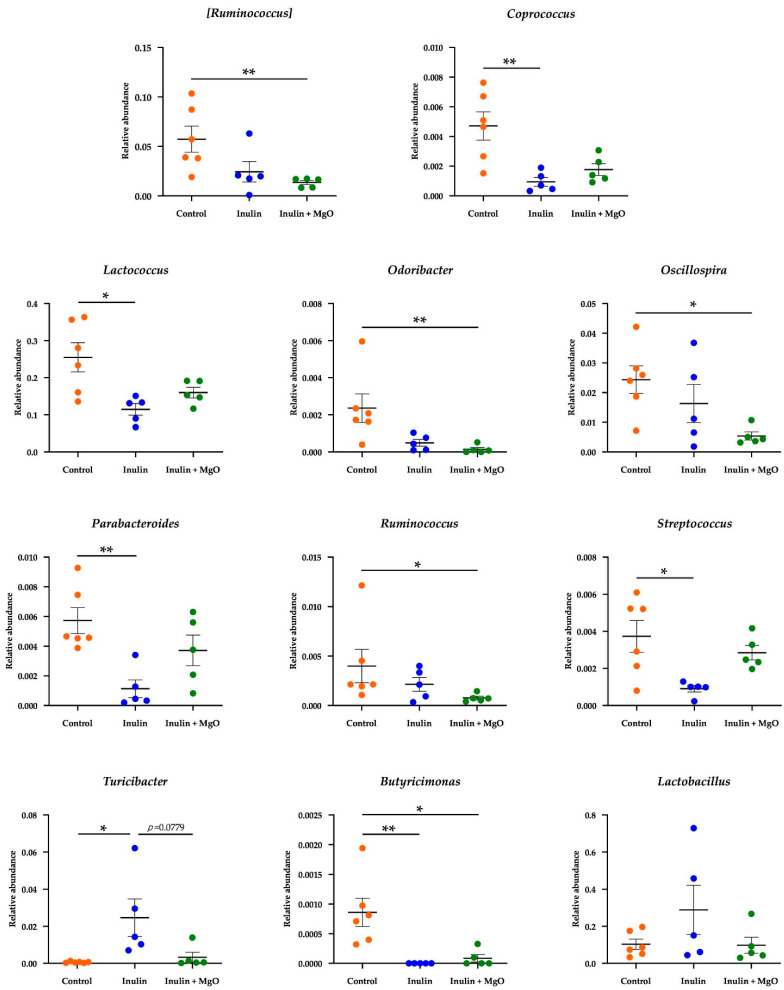
The combined inulin and MgO effects on intestinal bacteria at the genus level in mice on a HFD. Inulin was mixed with the HFD at a ratio of 2.5% in the inulin group and the MgO combination group. Magnesium oxide was given in the diet at a ratio of 0.25%. All data are expressed as mean ± standard error (Each group *n* = 5–6). * *p* < 0.05, ** *p* < 0.01. Data were evaluated using the Kruskal-Wallis test with Dunn’s post hoc test. MgO: magnesium oxide. HFD: high-fat diet.

**Figure 7 nutrients-13-00152-f007:**
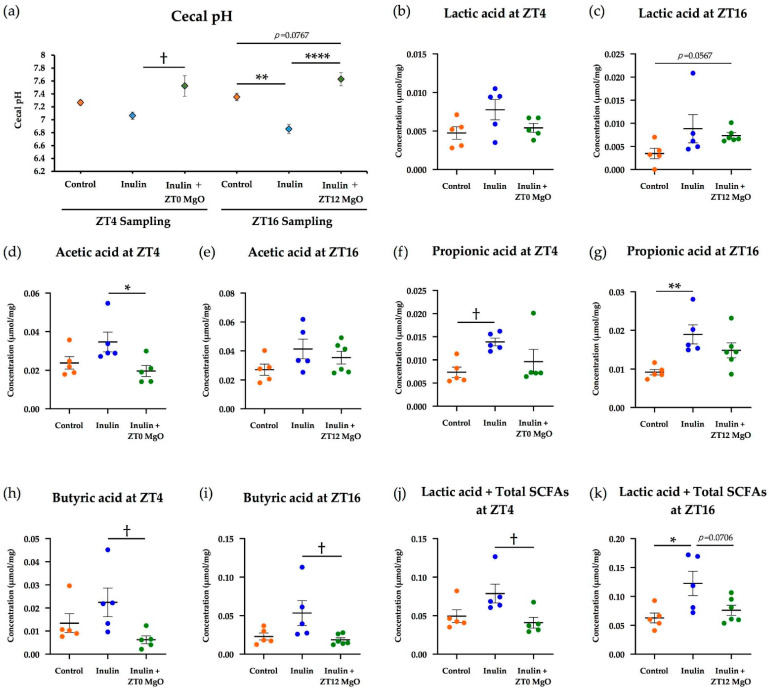
The MgO administration timing effects on cecal pH and cecal SCFAs in mice. (**a**) Cecal pH. (**b**) Lactic acid concentration of the cecal contents at ZT4. (**c**) Lactic acid concentration of the cecal contents at ZT16. (**d**) Acetic acid concentration of the cecal contents at ZT4. (**e**) Acetic acid concentration of the cecal contents at ZT16. (**f**) Propionic acid concentration of the cecal contents at ZT4. (**g**) Propionic acid concentration of the cecal contents at ZT16. (**h**) Butyric acid concentration of the cecal contents at ZT4. (**i**) Butyric acid concentration of the cecal contents at ZT16. (**j**) Lactic acid concentration and total SCFA concentrations of the cecal contents at ZT4. (**k**) Lactic acid concentration and total SCFA concentrations of the cecal contents at ZT16. Total SCFA concentration is the total of acetic acid concentration, propionic acid concentration, and butyric acid concentration. Inulin was mixed with the HFD at a ratio of 2.5% in the inulin group and the MgO combination group. Magnesium oxide at 250 mg/kg was orally administered to the inulin + MgO group. The MgO administration time was ZT0 or ZT12. Concentrations of lactic acid and each SCFA were calculated per dry weight of the cecal contents. All data are expressed as mean ± standard error (Each group *n* = 5–6). * *p* < 0.05, ** *p* < 0.01, and **** *p* < 0.0001, evaluated using the one-way ANOVA with Tukey’s post hoc test. ^†^
*p* < 0.05, evaluated using the Kruskal-Wallis test with Dunn’s post hoc test. MgO: magnesium oxide. SCFA: short-chain fatty acid.

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
