# Peer review of "The Combined Effects of Magnesium Oxide and Inulin on Intestinal Microbiota and Cecal Short-Chain Fatty Acids"

_nutrients, 2021, doi:10.3390/nu13010152_

Round 1

Reviewer 1 Report

The study is an important contribution on the understanding on how magnesium and inulin work in modulating constipation.  The authors addressed the problem with the right experimental design.

A minor comment and suggestion is that the Discussion section be segmented or broken down into sections with sub-headings to make it easier to read.

Author Response

We attached file

Reviewer 2 Report

This study aimed to investigate the MgO effects on SCFA and lactic acid concentrations in the cecum content and on the fecal microbiota diversity when the water-soluble dietary fiber, inulin. The appropriate MgO administration timing are also examined.

Major issues

From the introduction, we got the information that MgO and dietary fiber, inuline, are both recommended  as treatments for relieving constipation. But the experiments in this study use normal mice and high fat diet models to study the effect of MgO and/or inuline on cecal environment. That raises many questions. Could the results here be used to interpret conditions in constipation? Will the combination of MgO and inuline have synergistic or antagonistic effects on relieving constipation?

Minor issues

Materials and methods

2.2.1 Experimental design

The description on experimental design is not clear enough that reader can't easily understand how may groups and treatments were arranged in an experiment until we check the results.

In experiment 2 high-fat diet was used to interpret deteriorated intestinal environment. No normal control or parameters that could be used to demonstrate that the model was well established as expected. 

Line 184 2.2.5.16. S rDNA Sequencing should be 2.2.5. 16S rDNA Sequencing

In section 2.2.6 Analysis of 16S rDNA Gene Sequence, all "reads" are mistyped as "leads".

Line 232, "There were no significant differences in cecal pH between the two groups". There are 4 groups in this experiment, why only "two groups" are mentioned?

Discussion

Results are repetitive described in this section. 

Author Response

We attached file.

Reviewer 3 Report

The effects of inulin administration alone and in combination with Magnesium Oxide on gut pH, SCFA composition and microbial community composition in mice, are presented in this manuscript. It is a nice, well described study with interesting results.

Minor Points:

  1. Line 63: This is a misinterpretation of the findings from, Qin et al. Nature 2010, “A human gut microbial gene catalogue established by metagenomics sequencing”, propagated from reference 10. The original citation would be more appropriate to use here and the sentence should be updated to reflect the findings from that study.
  2. Line 70 – 73: A reference is needed for the information in these lines.
  3. Line 78: The statement that SCFAs “reduce the risk of obesity”, is a bit overstated given the current evidence. There is an association, but a causal link has not been definitively identified as far as I am aware.
  4. Line 103: The acronym, ICR should be defined.
  5. Line 211 – 219: I believe the word “leads” should be replaced with “reads” or “sequences” throughout this paragraph.
  6. Line 221 – 228: Here and in figure 4; were p-values corrected for multiple comparisons?
  7. Figure 3: Because mice are coprophagic, the mice that share a cage, often share a similar microbiota, which can be very different from other cages in the same experimental group (often referred to as “cage effect”). How many mice were housed together per cage in this experiment? This information should be included in the methods.
  8. Figure 4: It’s hard to tell which abundance values represent the taxa in the legend. It might be helpful to group very low abundant taxa into a single group called “other” or “low abundant taxa” to make the figure easier to read.

Author Response

We attached file.

Reviewer 4 Report

The work has interesting elements for the scientific community,  I believe that the conclusions may be useful for other researchers who carry out a line of research relating to constipation/SCFAs. Unfortunately, there are some points to be fixed in my opinion.

  • The M&M 2.1 lacks the number of mice used for the experiments. Link to that missing information I suggest using a different graph (Boxplot/Dotplot instead of histograms) to show the results.
  • I did not find the link/code for download the sequencing output used in this work. You have to deposit the reads somewhere (ENA sequencing, NCBI platform...) in order to be available for the scientific community.
  • I fig1(e) I found it strange that the Inulin +0.25MgO was not significantly different from the control.
  • Line 288: “Although the difference was not significant, the combined use with MgO reduced the lactic acid concentration (Figure 2b)” This is not possible to say like that. If you want to give a message like that you must at least put the p.value even if it is not
  • How do you explain the different pH values between the two experiments (“normal diet” mice and mice on HFD)? Could be interesting to put a few rows about that in the manuscript
  • If you want to show the alpha diversity you have to insert at least one index for evenness and richness characteristics. For instance: Simpson index (evenness index) and others like PD, observed otus, CHAO1 (richness level indexes) (Figure 5).
  • Concerning the SCFAs and lactic acid measurement, how you could measure the concentration without a MS analysis or the derivatization process?
  • In the discussion you give a lot of information, some of them are not strictly related to this work. I suggest reducing focusing on the “story” that you would like to communicate with this paper.
  • In line 69 "in vivo" should be italicized and not underlined.
  • In line 213 I think you mean Greengenes database.
  • In line 216: You should change “taxonomy “with “taxonomic composition”.
  • In line 217: change “UniFrance” with “UniFrac”.

Author Response

We attached file.

Round 2

Reviewer 2 Report

The revised version is clearly presented the design and results for publication.

Minor comments as below:

  1. Line 516 & 517, Our results are consistent with the results of a previous study observed in the small intestine when rabeprazole or vonoprazan was administered. Please cite reference.
  2. Line 517-519, These results suggest that MgO may strongly or over a long time lower the intestinal pH through its antacid action in the stomach, and affect the intestinal flora more strongly than PPI administration. The "lower" should be "increase" according to the content.

Author Response

We attached and uploaded response letter. 
